# The Role of Stomach Infrastructures on Children’s Work and Child Labour in Africa: Systematic Review

**DOI:** 10.3390/ijerph18168563

**Published:** 2021-08-13

**Authors:** Dagim Dawit Gonsamo, Herman Hay Ming Lo, Ko Ling Chan

**Affiliations:** Department of Applied Social Sciences, The Hong Kong Polytechnic University, Kowloon, Hong Kong; dagim.gonsamo@connect.polyu.hk (D.D.G.); herman.lo@polyu.edu.hk (H.H.M.L.)

**Keywords:** child labour, social transfer, social protection, child-sensitive, cash transfer

## Abstract

Child labour remains a prevalent global concern, and progress toward eradicating harmful children’s work appears to have stalled in the African continent and henceforth, integrated social policy intervention is still required to address the problem. Among several forms of social policy interventions, stomach infrastructure (i.e., in-kind and/or cash transfers) have been a key policy approach to support vulnerable families to lighten households’ resources burden, which forces them to consider child labour as a coping strategy. There is growing evidence on the impacts of these programs in child labour. However, this evidence is often mixed regarding children’s work outcomes, and the existing studies hardly describe such heterogeneous outcomes from the child-sensitive approach. To this end, a systematic literature search was conducted for studies in African countries. From 743 references retrieved in this study, 27 studies were included for the review, and a narrative approach has been employed to analyse extracted evidence. Results from the current study also demonstrate a mixed effect of in-kind and cash transfers for poor households on child labour decisions. Hence, the finding from the current review also demonstrates a reduced participation of children in paid and unpaid work outside the household due to in-kind and cash transfers to poor households, but children’s time spent in economic and non-economic household labour and farm and non-farm labour, which are detrimental to child health and schooling, has been reported increasing due to the program interventions. The question remains how these programs can effectively consider child-specific and household-related key characteristics. To this end, a child-sensitive social protection perspective has been applied in this study to explain these mixed outcomes to inform policy design.

## 1. Introduction

Child labour is described as work that deprives children’s potential, dignity, and childhood, is detrimental to physical and mental development, and interferes with schooling [1]. It remains a prevalent social problem in low- and middle-income countries [2]. According to the latest global estimates, 160 million children—63 million girls and 97 million boys—were in child labour at the beginning of 2020, accounting for almost 1 in 10 of all children worldwide. The same sources report that global progress against child labour has stagnated since 2016, and the percentage of children in child labour remained unchanged over the four-year period while the absolute number of children in child labour increased by over 8 million. Similarly, the percentage of children in hazardous work was almost unchanged, but rose in absolute terms by 6.5 million children [3].

In Asia and the Pacific, Latin America, and the Caribbean, child labour has curved down over the last four years in percentage and absolute terms. However, in the Sub-Saharan African region, an increase in both the number and percentage of children in child labour has been recorded since 2012. There are now more children in child labour in Sub-Saharan Africa than in the rest of the world combined, where 86.6 million (23.9%) are child labourers in this sub-region. To this end, global child labour goals will not be achieved without a breakthrough in this region [3].

While ending child labour is one of the key targets of sustainable development Goal 8.7, evidence suggests that the progress toward ending child labour by 2025 is still insufficient to meet the target and, henceforth, integrated social protection investment is a key to achieve the target [4].

Hence, social protection programmes have been increasingly recognised as a key strategy for reducing poverty and vulnerability [5]. However, only 35 per cent of global children enjoy effective access to social protection, whereas almost two-thirds of children are not covered with any forms of social protection; most of these children are from Africa and Asia [6], which suggests the need to increase social protection coverage for most vulnerable children in these parts of the world. Additionally, in most parts of Sub-Saharan African countries, children’s work is a normal part of their development, and a useful component of their everyday socialisation, sources of livelihood, schooling, and social relationships. To this end, it is most challenging to draw a strict boundary between children’s work and child labour, as children’s participation in economic activities in the African context, in general, is alleged to be useful for children’s well-being [7], and yet conceptualisation of harms on children’s lives rarely incorporate children’s and parents’ perspective. In the context of Africa, in particular, vulnerability is multidimensional, and childhood is not a time free from responsibility; hence, many children continue to make economic contributions to their households through their work while also attending schools [8,9].

On the other hand, stomach infrastructure in the form of social protection support for vulnerable and poor families has been a key policy instrument that has been implemented in most parts of African countries to support poor families to reduce their reliance on child labour as a coping strategy. Hence, the effects of these programs on child labour decisions have been widely recognised in the existing literature [10,11,12]. However, these studies on the effects of in-kind and cash transfer to vulnerable families rarely incorporate child-specific and household-related factors which determine social policy effects on child labour decisions. In addition, in the context of Africa, as child work is also the result of household-specific, school related [13], and community factors, examining whether such interventions have given emphasis to such interconnected factors is an important question to ponder.

More specifically, in the context of Africa, where child labour is highly prevalent and social protection coverage is limited, benefit levels are insufficient [6], and child work is intersecting with schoolings and children’s growth [8], a context-specific inquiry is essential to understand how child-specific and household-related factors determine the effects of cash or in-kind transfer on household and children’s decision on participation in economic activities.

## 2. Literature Review

Child labour is a complex phenomenon resulting from the individual, community and societal levels risk factors influencing household’s decisions [10]. Parents’ decisions regarding the work and schooling of their children are influenced by factors at the household, societal and community levels and characteristics of the context in which the household is living [14]. These include socio-demographic and economic factors such as poverty, neglect, lack of adequate care, exposure of children to various grades of violence, parental education status, gender, place of residence, household size, residence type or size, wealth index, parental survivorship, and household size [15].

Though household poverty is a key factor for child labour, evidence suggests that increases in household income might not necessarily result in significant reductions in child labour unless other possible risk factors such as structural, geographic, cultural, seasonal and school-supply factors, as well as gender and other demographic traits, are equally addressed [16]. Furthermore, given that these children are required to meet their basic need and costs of schooling, loss of income caused by removing children from work leave them worse off, and caused them to be involved in a work that could be even hazardous and interfere with school and other activities [10,17,18]. Likewise, an attempt to ban child labour through enforcement of minimum employment age also could not yield desired results for all children, as most working children in Africa are involved in agriculture and informal sectors where such mechanism is less likely to be effective [19].

In response to children’s vulnerability, social protection has been a policy option and has become increasingly prominent [20]. It is a set of actions implemented by the state, which aims to support individuals and families in dealing with vulnerabilities throughout their life-cycle, and especially help the poor and vulnerable groups become more resilient against crises and shocks [21]. Social protection programs are a mix of contributory and non-contributory schemes. Contributory social assistances are program types wherein contributions are made by targeted beneficiaries (and their employers) for entitlement to benefits (e.g., social insurance) and on the other and non-contributory programmes targeted towards the poor, and it covers only those people whose assets or income fall under a certain threshold [6].

Additionally, non-contributory transfers are either based on conditions or without conditions. Transfers with conditions are where beneficiaries are required to comply with some rules or expected behaviours to receive the transfer (usually related to school attendance or health care). On the other hand, unconditional transfers are financial or in-kind transfers for disadvantaged people without requiring anything in return or conditions to receiving benefits (e.g., unconditional child grant) [20,22,23]. Moreover, poverty-targeted cash transfers have been deemed to be social protection instruments that are becoming increasingly popular in low- and middle-income countries [24].

Though there is a growing body of empirical evidence suggesting the positive impacts of social transfer programmes on child well-being in general and reducing child labour, given the multiple risks that children, their families, and the community face, whether social transfer programs have reduced and/or increased child labour is an empirical question that needs to be investigated; especially considering children’s attributes (age, gender, agency, and child’s preference to work); intra-household dynamics (gender of caregivers and household size), and other interrelated factors, the question remains whether the social transfer programs reduced children’s vulnerability to child labour, and how these programs can work best to produce a positive outcome for the well-being of children in adversity.

The previous reviews [10,11,25,26,27], primarily report impacts of income or resources transfer to families and children on child labour decision in terms of income poverty and these studies rarely addressed child-specific and household-related factors through a child-sensitive social protection perspective, which addresses the multidimensional vulnerability of children. Hence, the current review applied a child-sensitive approach [28] which emphasises children’s multidimensional vulnerability: age, sex, location, agency, intra-household dynamics, reducing risks, and other related factors, to investigate the role of social transfers (cash or in-kind) to households and children in reducing child labour in the context of African countries.

The key contribution of the current review is to extend the existing knowledge on the positive role of social transfer programs and emphasise the possible ways by which such intervention can be more child-sensitive to genuinely reduce child labour and/or intensive child work and effectively address children’s context of work.

## 3. Methods

### 3.1. Search Strategy

A systematic search of English language peer-reviewed articles, impact evaluation reports, and grey literature through electronic databases and hand searching of organisation’s websites was conducted to find studies that examined the role of social protection interventions in child labour in African countries. The search was conducted between February 2021 to March 2021 using a combination of keywords: (“social policy” OR “cash transfer” OR “social program”” OR “social protection” OR “safety nets” OR “social subsidy”) AND (“child labour” OR “child work” OR “child labour”).

The following databases were searched for a full-text peer-reviewed journal article, review papers, official impact evaluation reports. Databases searched include PubMed; ScienceDirect; Web of Sciences; Scopus; ProQuest (PsycINFO; Sociological Abstracts; Social Services Abstracts; Social Work Abstracts); Cochrane Library; and Google Scholar. In addition, searches on other grey literature from relevant non-governmental and international organisation’s websites, such as UNICEF Office of the Research, World Labour Organization, The World Bank Development Impact Evaluation Initiative (DIME), World Bank Group e-library, and International Labour Organization (ILO), the Transfer Project were undertaken. The search was limited to studies published in the English Language between 2005 to March 2021. Reference lists of recent reviews and individual studies conducted on a similar theme were hand searched. Finally, the search resulted in a total of 743 references, which were imported into reference management tools for further screening. (Details of search terms and the search strategy for included database are shown in Table A1 in the Appendix A.)

### 3.2. Selection of Study and Inclusion Criteria

Search results were imported into EndNote 20 references management tool, and duplicates were removed electronically. Title and abstracts were reviewed to determine if the references imported might fulfil the inclusion criteria. After shortlisting, screened full texts were reviewed to check their characteristics for a final list of studies for inclusion. Thus, quantitative, qualitative, and mixed methods studies which deal with different forms of social protection intervention (such as cash transfer, school subsidy, food transfer, child grant, and other social support) having an impact on child labour and/or child work through addressing household poverty and vulnerability were included in this review. Therefore, these references were included in the current review if they met the following criteria listed in Table 1 below.

### 3.3. Data Extraction and Evidence Synthesis

In this review, evidence was only extracted from studies reporting the impact of social protection programs on child labour and child work from African countries, and data extraction form was used to extract data on the following components of studies included in the review: (1) country of study, (2) study design, (3) forms of social transfer program, and (4) program effect on child labour and/or child work outcomes, such as participation in any economic activities; household chores; working in farm and livestock herding; engagement hazardous activities; working in the family non-farm business; participation and time spent in domestic work outside the household; and employment in paid work outside the household. Moreover, to examine the child sensitivity of the social transfer programs, demographic profile (sex, age, etc.) of working children most reported as affected by the program in terms of child labour outcome and household-related factors, which explains heterogeneity in outcomes, are extracted. Hence, after relevant evidence was extracted, a narrative synthesis approach was used to organise the data on the key outcome variables.

### 3.4. Assessment of Risk of Bias in Included Studies

In this review, assessment of risks of bias for included studies was conducted using a tool developed by the International Development Coordinating Group (IDCG) secretariat to assess the risk of bias where the assessment focused on five categories such as selection bias and confounding, spill overs/crossovers/contamination, outcome reporting, analysis reporting, and other risks of bias [29]. This tool has been developed to assess the risk of bias for a range of quasi-experimental studies, as well as experimental studies [29,30], and has been used as a tool to assess the risk of bias on a previous study on a similar theme [31]. These risk assessment criteria were coded into three evaluation categories by [31], in which studies in which these items are clearly addressed were evaluated as “YES”, and those which failed each of the five criteria were evaluated with “NO”, and otherwise, if not stated, the “Unclear” label is assigned. Then, the overall risk of bias was aggregated as low, medium, or high, based on aggregation across the five categories given in Table A2 below. Hence, Low risk of bias: if ‘Yes’ for four or five categories; Medium risk of bias: if ‘Yes’ for three categories; and (3) High risk of bias: if ‘Yes’ only for two or fewer categories. Similar approach for assessment of risk of bias was applied in the current study.

### 3.5. Limitations of the Review

As the current review only included English language reports, the study has limitations in describing all contexts of social transfer programs in the continent (Africa) where there is substantial evidence with languages other than English, such as French, Portuguese, Spanish, and Swahili. The second limitation of the review is related to the data synthesis approach. Moreover, due to variation in the definition of child labour and outcome measurement across studies included in this review, it is not possible in this review to conduct a meta-analysis to estimate the pooled effect of social transfer intervention (cash or in-kind) on child labour outcomes. Hence, the study mainly used a descriptive approach to summarise studies reporting effects of social transfer programs on commonly reported child work outcomes to illustrate the mixed evidence across the literature, and illustrate the importance of a child sensitivity approach in social transfer programs in reducing child labour and/or intensive child work.

## 4. Results

### 4.1. Search Result

A total of 743 references were obtained through a database search, from which 69 full-text reports were screened for eligibility and, finally, a total of 27 studies met the inclusion criteria and were included for this review (Figure 1). In addition, as shown in Table 2 below, from 27 studies included in this review, 63% (n = 17) are peer-reviewed journals articles; 26% (n = 7) are impact evaluation reports, and 11% (n = 3) are working papers which employ various study designs including quantitative, qualitative, and mixed study design.

### 4.2. Program Type and Study Design

As discussed in the above section, the social protection program includes a range of program types with diverse beneficiary groups. The current review however only focused on studies on stomach infrastructures in the form of cash and/or in-kind transfers to poor families. Hence, as Table 2 below indicates, nearly all included studies (n = 23) are on cash transfer programs, whereas the remaining four studies are in-kind transfer programs. These studies also fall under the category of either conditional and/or unconditional transfer programs.

Regarding the study design, studies included in this review adopted quantitative, qualitative, and mixed study approach. As illustrated in Table 2, two-thirds of the studies (77.7%; n = 21) adopted a purely quantitative approach, while only three studies (11%) employed a purely qualitative approach, and three studies (11%) used a mixed approach. These figures indicate that most of the studies examining the impact of the social transfer on child labour outcome use a quantitative approach such as Randomized Control Trial (RCT) to examine the effectiveness of social transfer programs. However, despite its unique advantage to examine the effectiveness of an intervention (i.e., social transfer program) in the context where child labour is the result of complex individual, family, and societal level factors, examining experiences and perception of the beneficiaries also provide a rich understanding of the social, economic, and demographic context in which the social transfer programs might not work best, or have less impact.

In this regard, most studies on the impact of social protection programs tend to adopt the quantitative design, opinions and experiences of beneficiaries and the community are rarely documented in such studies [32]. Further the qualitative approach helps to better understand how and why the change occurred (or did not occur) [33].

Examining experiences and beneficiary views on the program’s effect on child labour requires the use of a community-based mixed approach to further understand the role of social transfer programs on the lives of the community. Hence, in addition to the apparent advantages of quantitative approaches to evaluate program effectiveness, a qualitative approach allows an in-depth understanding of the context and mechanisms through which a program may generate a different outcome in different circumstances, and can provide rich data with a more grounded understanding of the various pathways to inform child-sensitive social protection programming and policies.

### 4.3. Note: UCT: Unconditional Cash Transfer; RCT: Randomized Control Tria the Role of Stomach Infrastructure on Child Labour

This section summarises results reported in studies included in the current study on the impact of social transfer program on child labour and children’s work outcomes. More specifically, the current study reported effects on the commonly reported outcomes across child labour studies, such as (i) participation in any economic activities; (ii) participation and time spent in household chores; (iii) working in the farm and livestock herding; (iv) engagement in hazardous activities; (v) working in the family non-farm business; (vi) participation and time spent in domestic work outside the household (either paid or unpaid); and (vii) employment in other paid work outside the household. Thus, data were extracted on these outcomes from included studies and results are reported in the current study.

Table 3 below illustrates the effects of social transfer programs (cash and/or in-kind) to families and children on child labour and children’s work outcomes. As the summary of the evidence reported in Table 3 illustrates, the impact of cash and/in-kind transfer has a diverse impact on child labour and child work which interferes with their learning in school. Among studies included in the current review as illustrated in Table 3 below, seven (n = 7; 25.9%) studies [34,35,36,37,38,39,40] reported increased participation of children in any economic activities in general due to the social transfer programs, whereas four (n = 4; 14.81%) studies [41,42,43,44] reported a reduction in children involved in these activities.

Regarding children’s involvement and time spent in household chores, ten (n = 10; 37.03%) studies [34,35,36,38] reported increased participation of children in household chores due to social transfer to their families, whereas only three [39,44,45] studies (n = 3; 11.11%) reported a reduction in child working time in the household chores; in the latter, only one study [45] reported a significant result of a reduction in minutes a child spent per day (−22.082 * minutes for boys; and −48.658 ** minutes for girls) working in the household chores.

The third outcome reported in studies included are related to children’s involvement and time spent working in farm and livestock herding. Among studies included in the review which reported both positive and negative outcomes regarding children’s involvement and time spent in farm work and livestock herding, only three studies [44,45,46] reported a decrease in children’s involvement in farm and livestock herding due to the transfer, and three other studies [35,38,47] on the contrary reported increasing participation of children and more time spent in farm labour, by which both results are significant. With regard to the fourth outcome of measure (involvement in hazardous activities); two studies [34,38] reported children’s exposure to dust, fumes, or gas, and exposure to extreme heat, cold, or humidity (0.115 *** for boys and 0.090 *** for girls) and engagement in hazardous productive activities (both gender 0.044 **), respectively. Furthermore, related to the fifth outcome (children’s involvement in family non-farm business); three studies [39,43,48] reported a decline in children’s involvement in non-farm business due to the transfer, whereas two studies [38,47] reported a significant increase in time spent working in own family business.

Finally, with regards to paid work outside the household, among 27 studies included in the review, twelve (n = 12; 44.4%) studies [33,37,38,43,48,49,50,51,52,53,54,55] reported a decline in decline in children’s involvement in paid work outside the household; from these, five studies [38,48,51,52,54] reported significant results. Additionally, one study reported significant decline in children’s involvement in paid domestic work outside the household is reduced by 7.7% [36](*p* < 0.000).

As Table 3 clearly depicts, most studies included in the review account a mixed outcome; in a single study, a given social transfer program reduced some form of child labour or child work on the one hand, but on the other, increases or shifts child labour to other forms of child work and child labour; mostly from outside labour to household intensive work.

On the contrary to the above scenarios, in which most studies reported some forms of child labour or child work-related outcomes, three studies included in this review [56,57,58] reported that social transfer programs did not have any impact either in reducing or increasing child labour and/or child work. This can implies that cash or in kind transfer to poor families do not always reduce child labour and children’s work This is illustrated in a study from the Zambian’s Multiple targeting grant which reported that child labour had been increased irrespective of the program intervention, which might imply that the social protection program might not necessarily reduce child labour [56], and social cash transfer in Lesotho did not change child participation in economic activities such as working in farm and livestock activities in the household [59]. Moreover, the study from Ghana also reported that the Livelihood Empowerment against Poverty (LEAP) program did not have a significant impact on children’s paid work outside the household [49].

## 5. Discussion

Results from the current review inform a mixed effect of in kind and cash transfer to vulnerable families on child labour/child work outcomes. In support of this finding, a review and meta-analysis by [26] on the impact of cash transfer on child labour found that cash transfer programmes reduced child labour by 7 per cent on average, yet reported that these findings were moderated by gender, and that the program reduced work participation for boys by 7 per cent, but had no significant impact for work undertaken by girls.

Moreover, the other review on the effect of unconditional cash transfer in low- and middle-income countries also reported uncertainty in the effect of unconditional cash transfer on the likelihood of children engaged in child labour [11]. Similar results were also presented in a review by [27] on Latin America and African countries, who identified contrasting effects of cash transfer on child labour. In this review, a study from Colombia reported a decline in the amount of time spent on work by the student due to program intervention; whereas a study from Malawi, on the other hand, reported a significant increase in child labour among students receiving cash transfer [34].

Thus, contrary to the conclusion by [12], who contends that the cash transfer program does not increase child labour, [60] his study in Bolivia provided evidence that shows the probability that such programs can lead to increases in child labour. On this point, the study from India on the safety net mechanism also found evidence indicating an increase in child labour as unintentional adverse effects of such social protection programs [61].

The other review in Sub-Saharan African countries by [25] also reported varied results of social transfer programs, such as decline in child labour due to social transfer program in Malawi and Kenya, limited impact of social transfer programs in Lesotho, Uganda, and Zimbabwe, and increase in child labour in terms of children’s participation in unpaid work in Zambia’s social transfer programs. One study from Zambia also reported an increase in child labour irrespective of the program, which shows that social transfer programs would not necessarily reduce child labour [56]. This was also reflected in a study [44] which reported that an increase in household wealth through a cash transfer does not necessarily lead to a decrease in child labour, and even an increase in school participation due to program intervention does not directly translate into child labour reduction, as children may end up being engaged in both. Likewise, the study from Burkina Faso also witnessed that programs that reduce both the time and the monetary costs of education are not necessarily sufficient to reduce child labour, even if they effectively increase school attendance [47].

The study from Malawi and Zambia on the impact of cash transfer programs also reported a mixed and inconclusive result, as it was found that the program had a positive contribution on children’s school attendance and material well-being on the one hand, and an increase in children’s engagement on works that may be detrimental to their health, such as activities that expose children to hazards in Malawi and excessive working hours in Zambia [38]. Additionally, a study from Ethiopia’s social cash transfer programme also reported a mixed result that, in rural areas, the transfer led to a half an hour reduction in the total number of hours children worked, while in urban areas, transfers had the opposite impact, worsening the child labour situation [44].

The possible explanation for such mixed reports in the literature regarding social protection intervention and child labour has a two-fold implication [32]. These includes on the one hand, many of the dimensions of children’s well-being often not heard or taken into account and on the other hand social transfer programs design insufficiently consider possible risk factors which may lead to adverse impacts for children such as increases in child work, domestic violence, inequalities, and/or the disruption of schooling or childcare arrangements [28]. 

Moreover, an approach that targets households with the assumption that all members of the household (including children) will benefit equally usually overlook children’s specific vulnerability and, thus, a range of components of social protection aimed at addressing the multidimensional vulnerability of children is required [40]. To this end, the child-sensitive social protection approach has been stressed as having the potential to address the dual needs of children by protecting them from risks and vulnerability and responding to their developmental needs.

### 5.1. Child Sensitive Approach to Mixed Outcome in Child Labour

In this review, near to half of the studies included reported the potential role of social transfer programs in reducing child labour. However, the results for the study also indicate mixed results in child labour outcomes due to social transfer programs. In general, such diverse outcomes can be due to program design and other child and household-specific factors that should be equally considered to obtain more improved results. Irrespective of the nature of the program design (conditional vs. unconditional transfers), transfer size, and other factors external to the children and their families, in this review, we considered individual child and household level factors explaining the possibility of heterogeneous impacts of social transfers on child labour that urge the need to apply a child-sensitive approach.

Hence, to examine such factors in detail, key principles of child-sensitive social protection [28] has been used to explain the possible reason for heterogeneous effects of the social transfer on child labour. The child-sensitive social protection approach is built on the assertion that children’s vulnerability is multidimensional, therefore recognising children’s context of work is equally critical [62]. The current review has summarised some of the following factors, possibly explaining the mixed outcome, and it is described in Table 4 as follows.

The above Table 4 illustrates child-specific and household-related factors which resulted in various impacts on social transfer programs on the decision of child labour. Studies sometimes report program design, transfer size, and implementation procedures as some of the key factors for the heterogeneous outcome of social transfers on a decision on child labour and/or child work. Additionally, despite individual attempts by single studies which reported the this, the has been no synthesised evidence reporting the heterogeneity or mixed results of social transfer program through a perspective of child-sensitive social protection. Hence, the following section illustrates child-specific and household-related factors commonly reported across studies on this matter.

#### 5.1.1. Children’s Age

Stomach infrastructure to poor families has been found to produce various effects for children within different age groups. A child-sensitive approach is stressed to describe such heterogeneity in the current study. The impacts of social transfer on child labour vary by the age group and gender of children. Among studies included in the current review, age-specific child labour outcome has been reported in most studies, and child labour has been reported increasing with an increase in child age [34,36,51,53]. This implies that older children are more likely to participate in child labour than younger children, which social transfer programs need to consider.

#### 5.1.2. Child Gender

Gender variation among children in the household are also an important factor for heterogeneity in child labour outcomes. Among eleven (n = 11) studies included in the current review which reported reduction in child participation in labour, four studies reported that in-kind and cash transfers reduced participation for boys [46,50,51] than for girls, and one study reported reduced participation of both boys and girls in labour due to transfer programs [45]. The remaining eight (n = 8) studies did not report gender-related variation on effects of social transfer programs on child labour. On the other hand, social transfer programs increased both boys’ and girls’ participation in economic activities [34,40,41], and increased participation of boys as opposed to girls [47]. Hence, such heterogeneous effects of stomach infrastructure remain unexplained in the existing literature. Thus, adopting a child-sensitive approach would acknowledge the socio-cultural context of children’s work, in addition to a mere increase in transfer size and ascribing conditions on transfers, so that desired results can be achieved for both gender category.

#### 5.1.3. Forms and Intensity of Work

The impact of a social transfer program also varies with the form of activities in which children are involved, such as farm work, child work outside the household (paid and unpaid), and household chores. Among studies reporting a reduction in child labour and child work, seven studies reported a decline in children’s involvement in work (either paid or unpaid) outside the household [33,45,50,51,52,53,63]; two studies reported a decline in child labour in household activities [42,58]; and two studies reported a decline in extensive child labour on farms due to social transfers [45,46]. However, in terms of child labour and child work forms, these lists are not exclusive, as a decrease in household activities, labour outside the household, and child work on the farm has been reported to a varying degree in most studies.

Moreover, studies reporting an increase in child labour and child work intensity also reported an increase in child work outside the household, intensive farm work, and working in the household. These also show that a decline in one form of child work also increased other forms. Social transfer programs have been found to contribute to the reduction in child work outside the household. However, additional income transferred to the household was invested in productive activities, which increased child labour demand and work intensity in the household, exposed them to hazards, and affected their schooling [34,44,47,54]. Social transfers also increased household investment in productive activities in Malawi, which increased children’s exposure to hazardous work which exposed them to dust, fumes, gas, extreme heat, cold, or humidity [34]. This implies that, in addition to transfer size and modality, which have been a key factors for heterogeneity in impacts of social transfer programs on children’s participation on labour, the current review also must stress the need to consider the nature and intensity of work that children would be involved in due to social transfers to vulnerable households, as these programs have been found to reduce a given form of child work while increasing children’s participation in other forms of works which are still detrimental to their health.

#### 5.1.4. Children’s Agency and Work

Children’s agency, which is conceptualised as the capacity to act on their own [64,65], and their decision to engage in work to earn money, are key factors determining participation in economic activities, though have been rarely documented in the literature as key determinants of child work. The current review also found that children’s decision to be involved in work, irrespective of its detrimental effect, has been reported in few studies. Despite resource transfer to their caregivers, boys increasingly participate in economic activities to finance their education [47]. Perceived opportunity cost of attending schools among boys has been found a determining factor for children’s decision for increased involvement in farm work, which suggests that a child’s role in the decision to involve in work is also a key factor that might also contribute to increased participation of children in labour [35].

#### 5.1.5. Gender of the Household Head

In addition to child-related factors (age, gender, and children’s agency), the gender of the household head has been reported with variation in child labour outcome after program support. As [36] reported, as female-headed households invest the transfer in productive activities, for children in female-headed, children’s participation in non-household labour reduced (−9%), and engagement in household chores increased by 15% and with 0.42 h spent in labour due to the income transfer program. Moreover, [44] also reported a similar effect of social transfer in a female-headed household that the transfer used to pursue productive opportunities increased child work time in the household. Therefore, the current study illustrates that addressing child labour decisions through resource transfer to vulnerable families has to give significant emphasis to the context in which the children live and decide to work.

## 6. Conclusions and Implication for Further Research

Social transfer programs have a potential role in reducing children’s work outside the household for pay. However, they could not remove children from labour altogether, as the transfer size is generally too small to make a big difference, and not enough to take children out of work entirely. In this paper, adopting the child-sensitive approach, which emphasises the child’s and household level of vulnerability, the role of stomach infrastructure through in-kind and cash transfer to vulnerable families has been a strategy to reduce child labour in African countries. The current study found that stomach infrastructure would be an effective policy strategy to reduce child labour if they could give sufficient attention to child-specific and household-related factors determining the effects of policy intervention. Consequently, in addition to commonly stated factors creating heterogeneous results in child labour outcomes, such as program design, targeting strategies, and transfer size, emphasis on child-specific and household-related factors equally play a substantial role in the pathway in which the social transfer programs can work effectively to address adverse child labour outcome.

Evidence from the current review suggests that child and household-specific factors such as age, gender, children’s agency, gender of the household head, and forms and intensity of work require considerable attention to achieve a positive outcome from the social transfer program. To this end, adopting a child-sensitive approach in designing and monitoring social transfer through context-specific and in-depth inquiry into children’s perspectives and household characteristics is an important pathway. Therefore, policymakers and program managers need to emphasise such factors, clarifying how and why social transfer programs would either reduce or increase child labour and intensive child work in different contexts.

Furthermore, the existing studies on the role of social transfer on child labour primarily report the economic impacts of increased household income as contributing factors for reducing child labour. However, as most of these studies adopt a quantitative measurement, they rarely involve the perception, expectation, and experiences of care-givers, children, and the community regarding the actual benefits of the transfer program regarding reducing children’s vulnerability into labour works. To this end, as the child-sensitive social protection approach considers the voices and perspectives of children and their care-givers, future studies on these issues should involve multiple perspectives to understand factors contributing to children’s vulnerability to child labour beyond the economic aspects. Moreover, the lack of standard measurement regarding child sensitivity of social protection should also be addressed by integrating child-sensitive social protection principles with a rights-based perspective.

## Figures and Tables

**Figure 1 ijerph-18-08563-f001:**
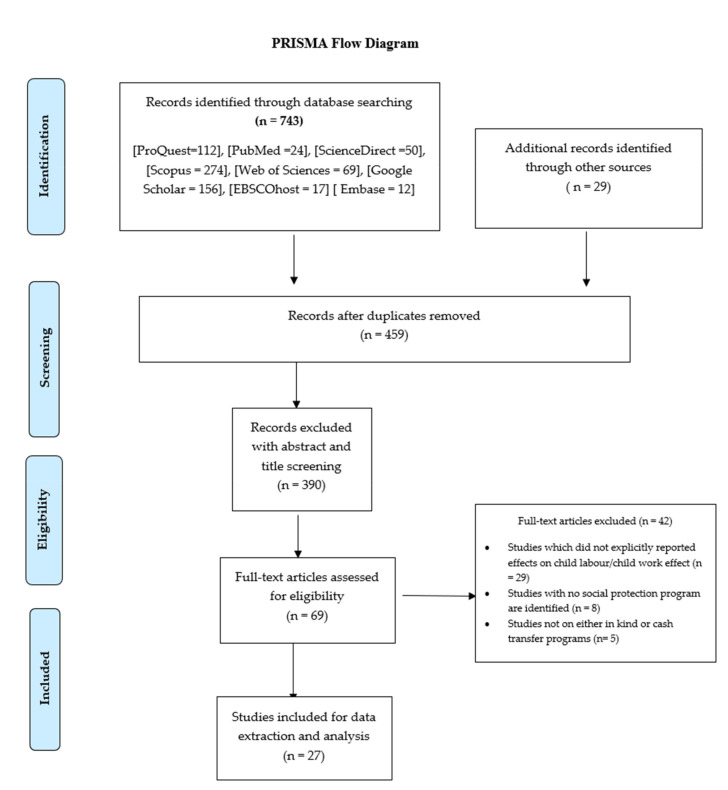
Prisma Flow Chart.

**Table 1 ijerph-18-08563-t001:** Study Selection Criteria.

Inclusion Criteria	Exclusion Criteria
Studies were included if they reported impacts of in-kind and or cash transfer on child labour and/or intensive child work	Studies were excluded if they did not report effects of cash or in-kind transfer).
Both qualitative and quantitative designs were included if they meet the above criteria	Studies on social transfer programs that did not report child labour outcomes (direct or indirect) were excluded.
As this study aims to describe Africa’s context, only studies conducted in African countries are included.	Studies that are not conducted at least in one of the African countries were excluded as this study intends to capture the African context alone.
	Commentaries, communication letters, conference abstracts, books, and book reviews were excluded

**Table 2 ijerph-18-08563-t002:** Characteristics of studies included (n = 27).

No	Study	Country	Social Protection Program Type	Type of Report	Study Design
1	Abdoulayi et al. (2016)	Malawi	Unconditional Cash Transfer (UCT)	Impact Evaluation	Mixed Design (Experimental, Interviews and group discussion)
2	AIR (2014)	Zambia	UCT	Impact Evaluation	Randomized Control Trail (RCT)
3	Asfaw et al. (2014)	Kenya	UCT	Journal	Randomized Control Trail (RCT)
4	Aurino et al.(2019)	Mali	In kind transfer(food)	Journal	Quasi-Experimental
5	Covarrubias et al. (2012)	Malawi	UCT	Journal	RCT
6	Daidone et al. (2014)	Zambia	UCT	Impact Evaluation	RCT
7	Jacobus De Hoop, Margaret W Gichane et al. (2020)	United Republic of Tanzania	Productive Safety Net Social Cash Transfer (Conditional and Unconditional)	Working Paper	RCT
8	J. De Hoop et al. (2020)	Malawi and Zambia	Social Cash Transfers	Journal	RCT
			Malawi (Unconditional)		
			Zambia (Multi Targeting)		
9	De Hoop and Rosati (2014b)	Burkina Faso	In-kind transfer (school kits, meals, and take-home rations conditional on school attendance)	Journal	Quantitative (Regression Discontinuity)
10	Dinku(2019)	Ethiopia	Cash and/or Food transfer (Conditional on public work	Journal	Longitudinal Survey
11	Fenton et al. (2016)	Zimbabwe	Social Cash Transfers (Conditional and Unconditional)	Journal	RCT
12	Fisher, Pozarny et al. (2017)	Malawi	UCT	Impact Evaluation	Qualitative Case Study
13	Handa et al. (2016)	Zambia	UCT	Journal	RCT
14	Kazianga et al. (2012)	Burkina Faso	In-kind transfer (food)	Journal	RCT
15	Kazianga et al. (2013)	Burkina Faso	In kind transfer (food)	Journal	Regression Discontinuity Design
16	Miller and Tsoka (2012)	Malawi	UCT	Journal	RCT
17	Fisher, Attah et al. (2017)	Kenya, Ethiopia, Malawi, Lesotho, Zimbabwe, and Ghana	Social Cash transfer	Journal	Qualitative Cross-Case Analysis
18	Owsu-Addo(2016)	Ghana	Conditional Cash Transfer (CTT)	Journal	Qualitative Descriptive
19	Pellerano et al. (2014)	Lesotho	UCT	Impact Evaluation	RCT
20	Pellerano et al. (2020)	Lesotho	UCT	Journal	RCT
21	Prifri et al. (2021)	Ethiopia	UCT	Journal	RCT Based Impact Evaluation
22	Angeles et al. (2017)	Ghana	UCT	Impact Evaluation	Quasi-Experimental
23	Sebastian et al. (2019)	Lesotho	UCT	Journal	RCT
24	Nanivaso(2013)	Malawi	Social cash transfer (Conditional and Unconditional)	Journal	Quantitative
25	Tafere and Woldehanna(2012)	Ethiopia	Cash and/or Food transfer (Conditional on public work	Working Paper	Mixed (Quantitative + Qualitative)
26	Rosas and Sabarwal (2016)	Sierra Leone	Cash and/or Food transfer (Conditional on public work	Working Paper	RCT
27	Evans et al. (2012)	Tanzania	Cash Transfers (Conditional)	Impact Evaluation	RCT

Note: UCT: Unconditional Cash Transfer; RCT: Randomized Control Tria.

**Table 3 ijerph-18-08563-t003:** Specific Outcome of Social Transfer on Forms of Child Labour and Work.

	Author	Child Age	Gender	Any Economic Activities	Household Chores	Farm Work and Livestock Herding	Hazardous Activities	Non-Farm Business	Domestic Work Outside	Paid Work Outside t
1	Abdoulayi et al. (2016)	6–17	boys	**0.088 *****	0.005 (excessive hrs)	NA	**0.115 *****	NA	NA	NA
			girls	**0.090 *****	0.013 (excessive hrs)	NA	**0.090 *****	NA	NA	NA
2	AIR (2014)		both	NA	NI	NI	NI	NI	NI	NA
3	Asfaw et al. (2014)	10–15	boys	NA	NA	**−0.120 **** (participation in farm work)	NA	NA	NA	NA
			girls	NA	NA	−0.072 (engagement in farm work)	NA	NA	NA	NA
4	Aurino et al.(2019)	7–16	boys	**0.142 *****	0.053 (participation)	**0.130 ***** (participation)	NA	NA	NA	NA
			girls	0.004	0.044 (participation)	−0.052 (participation)	NA	NA	NA	
5	Covarrubias et al. (2012)	–	both	0.003	**0.077 *****	0.021	NA	NA	**−0.077 *****	NA
6	Daidone et al. (2014)	5–18	boys	0.079 (unpaid)	NA	NA	NA	NA	NA	–0.017
			girls	0.002 (unpaid)	NA	NA	NA	NA	NA	–0.014
7	Jacobus De Hoop, Margaret W Gichane et al. (2020)	12–17	boys	–0.016	0.001 (both gender)	–0.008	NA	–0.007	NA	**−0.003 *****
			girls	0.004.		0.005 (excluding livestock)	NA	–0.002	NA	–0.008
8	J. De Hoop et al. (2020)	8–17 (Malawi)	both	0.034.	**0.091 *****	**0.063 **** (excluding livestoc)	**0.044 ****	0.002.	NA	**−0.061 *****
		8–17 (Zambia)		**0.055 ****	**0.031 ***	NA	NA	0.034 *	NA	0.001.
9	De Hoop and Rosati (2014b)	5–12	both	NA	**0.179 ***** (without female siblings); **0.179 ***** (with female siblings)	**0.063 **** (with female siblings); 0.029 (without female siblings)	NA	**0.095 ***** (with siblings); **0.089 ***** (no siblings)	NA	NA
		5–12	girls	NA	**0.052 ***	−0.027	NA	**0.082 ****	NA	NA
10	Dinku(2019)	-	-	NA	NA	NA	NA	NA	NA	**−0.107 *****
11	Fenton et al. (2016)		no variation	NA	NA	NA	NA	NA	NA	**−0.41 *****
12	Fisher, Pozarny et al. (2017)		no variation	NA	Increased	Increased	NA	NA	NA	**Reduced**
13	Handa et al. (2016)	11–14	no variation	0.0631	NA	NA	NA	NA	NA	**−0.0502**
14	Kazianga et al. (2012)	6–15	boys (meals program)	**0.064 ***	−0.021	NA	NA	−0.020	NA	NA
			girls (meals programs)	0.018	0.017	NA	NA	−0.011	NA	NA
			boys (take-home ration program)	**−0.064 ***	−0.066	NA	NA	**−0.091 ****	NA	NA
			girls (take home ration program)	0.018	0.076	NA	NA	**−0.115 ****	NA	NA
15	Kazianga et al. (2013)	6–12	no variation reported	NA	Increased	NA	NA	NA	NA	NA
16	Miller and Tsoka (2012)	6–18	boys	NA	**0.08 ***	0.09	NA	NA	NA	**−0.12 *****
			girls	NA	**0.11 *****	0.01	NA	NA	NA	**−0.10 *****
17	Fisher, Attah et al. (2017)	-	no variation reported	NA	NA	NA	NA	NA	NA	Reduced (“ganyu” labour)
18	Owsu-Addo(2016)	-	no variation reported	NA	NA	NA	NA	NA	NA	Reduced (Labour in illegal mining)
19	Pellerano et al. (2014)	6–17	boys	−7.29 (participation); and intensity of hr worked per week: **−3.156 ***	NA	−6.672	NA	-0.105	NA	−1.426
			girls	−2.014 participation; and (intensity of hr worked per week: −0.0227)	NA	−1.727	NA	0.465	NA	0.853
20	Prifri et al. (2021)	6–15	girls	–0.02 (hours worked −1.57); days of work: **−0.33 ***	NA	−0.02 (hours in the farm: **−1.80 ***) and days in the farm: **−0.37 ***	NA	NA	NA	NA
21	Prifri et al. (2021)	5–14	boys	NA	–0.213	−0.200(farm work); Livestock Herding: −0.057)	NA	0.053	NA	NA
			girls	NA	–0.400	–0.031(Farm work); Livestock Herding: −0.061	NA	−0.001	NA	NA
22	Angeles et al. (2017)	7–17	both	NA (NA: Not Available)	NA	NA	NA	NA	NA	−0.004
23	Sebastian et al. (2019)	13–17	boys	NA	**−22.082 *** min per day for all gender: and −1.092 min/day for boys	For all gender: −0.612 **; and for boys: **−0.824 **** (number of days worked in farm); and **−0.824 **** (number of days worked for boys)	NA	6.966 min/day	NA	NA
			girls	NA	**−48.658 ****	−0.337(number of days worked)	NA	−6.472 min/day	NA	NA
24	Nanivaso (2013)	6–18	boys	NA	NA	NA	NA	NA	NA	Reduced (28.4%)
25	Tafere and Woldehanna(2012)	7–17	both	**0.671 **** (hour spent/day)	**0.500 ***** (hour spent/day)	NA	NA	0.046 (hour spent/day)	NA	**0.314 *** (hour spent/day)
26	Rosas and Sabarwal (2016)	6–14	no variation reported	NI (NI refers: No Impact reported in the studies on the outcome of intrest)	NI	NI	NI	NI	NI	NI
27	Evans et al. (2012)	7–15	no variation reported	NI	NI	NI	NI	NI	NI	NI

**Note**: Results reported in **bold fare** outcomes reported as **significant** among included studies regarding different forms of child labour and/child work. * *p* < 0.1; ** *p* < 0.05; *** *p* < 0.01.

**Table 4 ijerph-18-08563-t004:** Child-specific and household factors explaining mixed outcome on child labour and/or work.

#	Study	Possible Reason Describing Mixed Outcomes
1	Abdoulayi et al. (2016)	Household Investment of the transfer on economic activities
2	AIR (2014)	Child Labour increased irrespective of the program intervention
3	Asfaw et al. (2014)	Proximity to to the local market places (increased distance from market reduced the likelihood of participation in paid work outside the household). Children’s living in near the local market are more likely to engage in child labour than those in
		Boys (0.048); Girls (−0.085)
4	Aurino et al. (2019)	Older boys’ preference to work. Perceived opportunity cost of schooling is higher among boys
5	Covarrubias et al. (2012)	Gender of household head:
		Children in female-headed households: 9% reduction in participation in paid domestic work outside the household); increased participation in household chores with 15.2%; and 4.24 h spent on household chores.
		However, for male-headed households, the transfer impacts are only in terms of reduced school absenteeism.
6	Daidone et al. (2014)	-
7	Jacobus De Hoop, Margaret W Gichane et al. (2020)	Gender, Age and Schooling:
		The reduction in child participation in paid work outside the household was significantly stronger for males than for females, for older children than for younger children, and for in-school children than for out-of-school children
8	J. De Hoop et al. (2020)	Child Age group:
		12–14 (participation in any economic activities or chores: 0.074 ****) and 15–17; 0.069 ** in Malawi. Excessive hours in economic activities or chores 0.046 * and 12–14 and 0.055 ** for 15–17 in Zambia
9	De Hoop and Rosati (2014b)	Gender dynamics in the HH (HH: Household)
		Boys with female siblings (participation in any economic activities or household chores: 0.078 **); (work in family business: 0.095 ***); (farming: 0.063 **).
		Boys without female siblings (any: 0.179 ***); (work in family business: 0.089 ***)
10	Dinku (2019)	Location (rural vs. urban).
		Gender of HH head (female-headed household: 0.078 ***), and child age (0.038 **);
11	Fenton et al. (2016)	NA
12	Fisher, Pozarny et al. (2017)	NA
13	Handa et al. (2016)	Child age 11–14 (−0.0502 *)
14	Kazianga et al. (2012)	NA
15	Kazianga et al. (2013)	NA
16	Miller and Tsoka (2012)	Child gender
17	Fisher, Attah et al. (2017)	NA
18	Owsu-Addo (2016)	NA
19	Pellerano et al. (2014)	Age group:
		6–12 (−1.643 *: intensity of hour spent in any labour activity during last 7 days)
20	Prifri et al. (2021)	Household size (0.37 **);
		Child gender: for girls (hours worked in the farm: 3.26 *; and days worked in the farm: 0.62 *)
21	Prifri et al. (2021)	Location: urban vs. rural environment.
		Gender of the household head
		Children in male-headed household: −0.102 ***); and
		Child age (between 5–14: −0.540 **)
22	Angeles et al. (2017)	NA
23	Sebastian et al. (2019)	Gender of the HH head: Girls in female-headed households spent 66 min/day (66.335 *) more time in doing chores than those in male-headed households
24	Nanivaso (2013)	Child gender: More boys move out of the labour market than girls due to the transfer (1.28%)
25	Tafere and Woldehanna (2012)	Children substitute for their parents or do work on their own.
26	Rosas and Sabarwal (2016)	NA
27	Evans et al. (2012)	NA

## Data Availability

Data is contained within articles included in the review.

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
