# Peer review of "The Role of Stomach Infrastructures on Children’s Work and Child Labour in Africa: Systematic Review"

_ijerph, 2021, doi:10.3390/ijerph18168563_

Round 1
Reviewer 1 Report
The study focused social transfer policy and child labour in Africa. It is an effort to address a major social problem affecting children in Africa.
Kindly consider changing the topic to "Stomach infrastructure and child labour in Africa: A systematic review"
The conceptualisation of child labour is poor/weak within the context of Africa. How do you describe children who work for survival even while attending school, and living with their parent as a public health issue? The concept of child labour within the context of Africa need to be address
The statistics use through out this study is dated for the subject under discussion
Please recast line 28-41 and be more objective using updated literature.
How can you describe a child who goes to school in the morning and support her mother in her school during the evening as child labour? Kindly interrogate the term Child work and child labour (See the work of Chukwudeh and Oduaran, 2021).
Child labour is not exclusive to outcome from household decision, what about external push factors
There has been studies on child labour that match your inclusion criteria and published by MDPI. Kindly consider their inclusion
The introduction is not sufficiently contextualise to suit the current argument of child labour in Africa.
There has been contemporary shift from discussion in the farm to urban space as related to child labour. Emphasis on work in the farm is an indication that the authors were reviewing dated materials.
Most of the references were not from African scholars that are affiliated to African institutions. How can foreign authors understand contextual issues more than African scholars? Kindly cite African scholars who are affiliated to institutions in Africa.
Kindly confirm the existence of these social programmes ascribe to each country on page 208-212
Author Response
Please see the attachment
Comments are written point by point with red.
Kindly

Reviewer 2 Report
This paper attempts to review the literature on the effects of government transfers on child labor, which is obviously an extremely important question. The paper does not claim novelty on reviewing the effect of government transfers on child labor.\footnote{references 7, 20-23, line 99} It claims novelty on reviewing the circumstances in which government transfers affect child labor. I find the paper lacks a little bit of focus: it spends more space confirming the results of former literature reviews and should focus more on the novel results. Besides, the main new results of the paper should be presented more clearly and are a bit disappointing overall. The subject of the paper is very interesting, it currently faces strong limitations, but I find there is much space to improve the paper. Besides, the current version of the paper failed to convince me that the literature is sufficient for a literature review on the heterogeneity of the effects of governments transfers on child labor.
(see the pdf for the detailed comments)

Author Response
Please see the attachment
Responses to comments are written in red.
Kindely
